# Comparative Study of Lesions Obtained through Radiofrequency between the Irrigated Ablation Catheter with a Flexible Tip and the Non-Irrigated Catheter in Ex Vivo Porcine Hearts

**DOI:** 10.3390/biology13020132

**Published:** 2024-02-19

**Authors:** Francesco Vitali, Martina De Raffele, Michele Malagù, Cristina Balla, Giorgia Azzolini, Federico Gibiino, Alberto Boccadoro, Marco Micillo, Matteo Bertini

**Affiliations:** Cardiology Unit, Sant’Anna University Hospital, Department of Translational Medicine, University of Ferrara, Via Aldo Moro 8, 44124 Ferrara, Italy; martina.deraffele@gmail.com (M.D.R.); bllcst@unife.it (C.B.); azzolini.giorgia@libero.it (G.A.); federico.gibiino@hotmail.it (F.G.); alberto.boccadoro@gmail.com (A.B.); marco.micillo942@gmail.com (M.M.); doc.matber@gmail.com (M.B.)

**Keywords:** radiofrequency, ablation, irrigated catheter, lesion characteristics, arrhythmia

## Abstract

**Simple Summary:**

Open-irrigated catheters create smaller, deeper lesions, while non-irrigated ones produce shallower lesions. This ex vivo study compares two different types of ablation catheters, inducing RF lesions on porcine myocardial slabs at three power settings. Irrigated catheters produce wider lesions, especially at 40 W, with a higher maximum depth. Non-irrigated catheters result in wider superficial areas, notably at 20 W. Irrigated RF ablation catheters may generate smaller superficial lesions than non-irrigated ones.

**Abstract:**

Background: At the same conditions of delivered power and contact force, open-irrigated radiofrequency ablation catheters are believed to create deeper lesions, while non-irrigated ones produce shallower lesions. This ex vivo study aims to directly compare the lesion dimensions and characteristics of an irrigated ablation catheter with a flexible tip and a non-irrigated solid-tip catheter. Methods: Radiofrequency lesions were induced on porcine myocardial slabs using both open-tip irrigated and non-irrigated standard 4 mm catheters at three power settings (20 W, 30 W, and 40 W), maintaining a fixed contact force of 10 gr. A lesion assessment was conducted including the lesion depth, depth at the maximum diameter, and lesion surface diameters, with the subsequent calculation of the lesion volume and area being undertaken. Results: Irrigated catheters produced lesions with significantly higher superficial widths at all power levels (3.8 vs. 4.4 mm at 20 W; 3.9 mm vs. 4.4 mm at 30 W; 3.8 mm vs. 4.5 mm at 40 W; *p* = 0.001, *p* = 0.019, *p* = 0.003, respectively). Non-irrigated catheters resulted in significantly higher superficial areas at all power levels (23 mm^2^ vs. 18 mm^2^ at 20 W; 25 mm^2^ vs. 19 mm^2^ at 30 W; 26 mm^2^ vs. 19 mm^2^ at 40 W; *p* = 0.001, *p* = 0.005, *p* = 0.001, respectively). Irrigated catheters showed significantly higher values of lesion maximum depth at 40 W (4.6 mm vs. 5.5 mm; *p* = 0.007), while non-irrigated catheters had a significantly higher calculated volume at 20 W (202 µL vs. 134 µL; *p* = 0.002). Conclusions: Radiofrequency ablation using an irrigated catheter with a flexible tip has the potential to generate smaller superficial lesion areas compared with those obtained using a non-irrigated catheter.

## 1. Introduction

Radiofrequency (RF) is the most common energy source for catheter ablation of cardiac arrhythmias. The main mechanism of lesion creation by RF ablation is that of heating the tissue. Lesion formation through the application of RF is contingent upon various parameters, including power, the contact force (CF) exerted between the catheter tip and tissue, the duration of energy delivery, the temperature, the tip size, and the tip orientation [1]. Using standard non-irrigated 4 mm tip ablation catheters, only a portion of the total power effectively reaches the tissue, while the remainder is dissipated in the blood pool and the circuit connecting the patient and the generator. Moreover, the absence of any residue on the electrode does not guarantee a safe ablation procedure as there remains a risk of steam pops [1,2]. Open-irrigated catheters can prevent blood clot formation, reduce the risk of steam pops [3], and have the ability to deliver a higher level of power to regions of reduced blood flow. Commonly employed in the treatment of prevalent supraventricular tachycardias, such as atrioventricular nodal reentrant tachycardia and right atrial flutter, standard non-irrigated 4 mm tip catheters exhibit variations in lesion size based on the superficial area, depth, and volume [1,4]. While older studies have reported that under similar conditions of delivered power and contact force, non-irrigated catheters tend to produce lesions with a shallower depth and irrigated catheters are associated with deeper lesions and higher volumes [1,2], there is a notable absence of direct comparisons between contemporary irrigated and non-irrigated catheters in a controlled environment with stable and reproducible contact forces, especially regarding the superficial lesion area and lesion volume. 

In addition to this, older ex vivo validation studies on non-irrigated catheters do not take into account blood flow and its cooling effects and often lack advanced contact force analysis. Moreover, open-irrigated catheters display distinctive characteristics in forming myocardial lesions, particularly when irrigation is targeted precisely at the tissue surface. In such instances, the resulting lesions may exhibit a further reduction in surface area while penetrating deeper into the tissue, with these results being attributed to the cooling effect induced by the irrigation on the myocardium surface [5]; however, few ex vivo validation data exist as yet. Understanding the differences in lesion formation and characteristics of contemporary irrigated and non-irrigated catheters is crucial to better understand which type of catheter is more appropriate for the treatment of different arrhythmias. In this study, we hypothesized that the flexible-tip open-irrigated ablation catheter would generate lesions characterized by a smaller surface area, deeper extension, the same volume, and a teardrop shape, directly comparing it with non-irrigated catheters that produce lesions with a larger surface area, shallower extension, and an elliptical shape.

## 2. Methods

### 2.1. Experiment Setup

Freshly excised pig hearts were used in the study, and the left ventricle was isolated by making a longitudinal incision to expose the endocardial surface. The tissue slice (7 cm × 3 cm) was then positioned on a platform suspended within a clear chamber, which was filled with saline solution (0.9% saline) and maintained at a temperature of 38 °C ± 0.5 °C. Using a manipulator, the ablation catheter tip was carefully placed horizontally to the surface of the myocardial preparation. To ensure stability, all catheters underwent testing with a stable tissue contact weight of 10 g, which was monitored through a high-precision digital scale (WJ-6000, Wunder^TM^, Milan, Italy). The catheter-to-tissue contact was constantly maintained during RF delivery through the real-time feedback provided by the digital scale. Inside the saline chamber, a pump was employed to generate non-pulsatile flow directed towards the ablation electrode–tissue interface at a rate of 6 L/min (Figure 1). The efficacy and validity of this system was assessed through a range of experiments [6,7]. In the first experiment [6], the system that was used aimed to evaluate the effects of catheter tip orientation on the superficial lesion length and the maximal lesion length of the myocardial lesion. In the second one [7], the system that was used aimed to validate a multiparametric index incorporating time as well as the power, contact force, and impedance recorded during ablation to assess lesion characteristics. The experiment was approved by our local ethics committee.

### 2.2. Ablation Catheters and the Radiofrequency Energy System

The study investigated the generation of myocardial lesions using two different ablation catheters as follows: an irrigated ablation catheter with a 4 mm laser-cut tip for uniform irrigation, an 8F shaft, and bidirectional handling (FlexAbility^TM^ Sensor Enabled Ablation Catheter, Abbott, St. Paul, MN, USA) and a non-irrigated catheter with a standard 4 mm solid-tip, non-irrigated, a 7F shaft, and unidirectional push–pull handling (Therapy^TM^ Ablation Catheter, Abbott, St. Paul, MN, USA). Both catheters featured a thermocouple embedded within the tip electrode to monitor the temperature during ablation.

In particular, the FlexAbility^TM^ (Abbott, St. Paul, MN, USA) catheter has a flexible tip achieved by laser cutting an interlocking dovetail pattern of kerfs on the cylindrical electrode’s surface. The variable expansion/compression of the kerfs in response to the axial or radial force allows the catheter tip to flex. When deflected, the kerfs direct the irrigation flow towards the tissue surface, reducing the overall flow and enhancing tip stability (Figure 2).

The ablation radiofrequency generator (Ampere^TM^ radiofrequency generator, Abbott, St. Paul, MN, USA) facilitated continuous digital and graphic monitoring of applied voltage, current, power (1 to 100 W), impedance (50 to 300 Ohm), and temperature (set at 60 °C for the 4 mm and 43 °C for the irrigated catheter). The current was applied in a unipolar mode between the distal electrode of the catheter and an indifferent lead placed underneath the myocardium, though not directly on it.

### 2.3. Ablation Protocol

Different endocardial lesions were created in different myocardial sections in a power control mode with a power of 20, 30, and 40 W for 60 s by 2 independent operators. Three ablation sessions were performed with the mentioned applied power for each of the two catheter types, for a total of six sessions. Ten lesions were performed on the surface of each ventricle slice. During each RF energy application, the delivered power, the impedance, and the temperature at the tip electrode were continuously recorded. The occurrence of an audible steam pop was noted and visually confirmed, and the lesion formed by steam pop was excluded from the count, but the number of steam pops in every session was collected.

### 2.4. Lesion Measurements and Characterization

Following the administration of radiofrequency energy, the myocardium was cross-sectioned at the level of each lesion (Figure 3). All lesions underwent examination for endocardial surface carbonization and crater formation, defined as the disruption of myocardial tissue. Carbonization manifested as superficial tissue damage with indications of endocardial surface burning. Using a digital precision caliper, we assessed lesion depth, (B) maximum length of the lesion at any depth, (C) depth at the maximum length, and (D) lesion surface diameters (length and width: D1 and D2) (Figure 4). We assessed superficial length (D1), defined as the long axis of the imprint left by the catheter tip in the tissue, and superficial width (D2) perpendicularly to the previous axis. Subsequently, the superficial area of each lesion was calculated (Figure 4). Additionally, each lesion volume was calculated, as previously described by Dorwarth et al. [1], considering the lesion as an oblate ellipsoid according to the following formula:V = [0.75π(B/2)2(A − C)] − [0.25π(D/2)2(A − 2C)]

### 2.5. Statistical Analysis

Continuous variables were expressed as mean ± standard deviation when normally distributed, as estimated using the Shapiro–Wilk test, or as the median and interquartile range. All variables were not normally distributed. Categorical variables were expressed as numbers and percentages.

Differences between repeated measurements were assessed using the Wilcoxon rank-sum test. Differences between independent measurement were assessed using the Mann–Whitney U test. *p* values < 0.05 were considered statistically significant. The statistical analysis was performed using STATA, version 15.0 (StataCorp LLC, College Station, TX, USA).

## 3. Results

A series of ablation lesions was conducted using the irrigated and non-irrigated ablation catheters in a power control mode, with power settings of 20, 30, and 40 W for 60 s. Three ablation sessions were performed for each catheter type, totaling six sessions. In total, 60 lesions were generated for both the irrigated flexible-tip and non-irrigated solid-tip catheters as follows: 20 at 20 W, 20 at 30 W, and 20 at 40 W (Table 1, Table 2 and Table 3, Figure 3).

Notably, no steam pops occurred with the irrigated ablation catheter regardless of increasing the power output. In contrast, four steam pops occurred with the non-irrigated catheter at 30 W, and eight ones occurred at 40 W. Lesion dimensions and characteristics are detailed in Table 1, Table 2 and Table 3, where the maximum lesion depth, maximum length of the lesion at any depth, depth at the maximum length, superficial length, and superficial width were measured for each lesion.

An irrigated ablation catheter determined lesions with significantly higher superficial widths at all power levels (3.8 vs. 4.4 mm at 20 W; 3.9 mm vs. 4.4 mm at 30 W; 3.8 mm vs. 4.5 mm at 40 W; *p* = 0.001, *p* = 0.019, *p* = 0.003, respectively) (Figure 5). Lesions performed using the non-irrigated catheter resulted in significantly wider superficial areas at all power levels (23 mm^2^ vs. 18 mm^2^ at 20 W; 25 mm^2^ vs. 19 mm^2^ at 30 W; 26 mm^2^ vs. 19 mm^2^ at 40 W; *p* = 0.001, *p* = 0.005, *p* = 0.001, respectively). An irrigated ablation catheter determined significantly hihger values in lesion maximum depth at 40 W (4.6 mm vs. 5.5 mm; *p* = 0.007), while no statistical difference emerged at lower power levels (4.2 mm vs. 4.2 mm at 20 W; 4.5 mm vs. 4.9 mm at 30 W; *p* = 0.428, *p* = 0.08, respectively) (Figure 5).

Lesions performed using the non-irrigated catheter had a significantly higher maximum length at 20 W (7.9 mm vs. 6.5 mm; *p* = 0.012), while no difference was observed at higher power levels (9.1 mm vs. 8.9 mm at 30 W; 8.6 mm vs. 9.4 mm at 40 W; *p* = 0.45, *p* = 0.141, respectively). The calculated volume of lesions performed using the non-irrigated catheter was significantly higher at 20 W (202 µL vs. 134 µL; *p* = 0.002), with no statistical difference observed at other power levels (231 µL vs. 227 µL at 30 W; 260 µL vs. 266 µL at 40 W; *p* = 0.86, *p* = 0.734, respectively) (Figure 5).

## 4. Discussion

In our study, the superficial lesion areas were greater with the non-irrigated catheter than with the irrigated catheter, irrespective of the power delivered. Furthermore, we observed that the irrigated catheter determined significantly higher values of lesion maximum depth when performed at a high-power delivery (4.6 mm vs. 5.5 mm at 40 W; *p* = 0.007) (Figure 5). Unexpectedly, the calculated volume of the lesions performed using the non-irrigated catheter was significantly higher at a low-power delivery (202 µL vs. 134 µL at 20 W; *p* = 0.002) (Figure 5). No steam pop was observed using the irrigated ablation catheter, while four steam pops occurred using the non-irrigated catheter delivering 30 W, and eight steam pops occurred when delivering 40 W.

Irrigated catheter tip technology has been introduced to improve the efficacy of complex catheter ablations without compromising procedural safety [8]. In the last few years, several “in vitro” and “ex vivo” studies have analyzed the parameters that might improve the procedural outcomes [9]. One of the first studies exploring lesions performed using irrigated ablation catheters was conducted in 2002 by Weiss et al. [10]. What emerged from this trial was that irrigated ablation prevented thrombus formation on the electrode surface, and smaller lesion diameters were created by increasing the irrigation flow rates [10]. Subsequently, Houmsse et al. demonstrated that the cooling of the tip causes a reduction in lesion size at the point of contact with the myocardium, thereby resulting in an equal or smaller surface lesion than that obtained using a non-irrigated tip [11]. Nevertheless, what emerges from both “in vitro” and “ex vivo” studies is that despite the progress that has been made in irrigation technologies, there is still a mild risk of complications such as thrombus formation or charring on both the catheter and tissue and steam pops with the potential risk of perforation in the myocardial wall [2,10]. Additionally, an important limitation is that of the maintenance of uniform contact during the delivery of RF energy with the tissue, which needs to be cooled. To overcome these limitations, an irrigated catheter with a flexible tip has been designed. The FlexAbility^TM^ catheter used in our study is one of the latest catheter tip designs proposed to obtain more uniform cooling of the catheter–tissue interface at low-perfusion volumes [12]. It utilizes a flexible tip generated through laser cutting an interlocking dovetail pattern of kerfs on the cylindrical sides of the electrode. The kerfs enable variable expansion and compression when axial or radial force is applied to the tip. When the catheter tip is deflected, the kerfs direct the irrigation flow towards the tissue surface, enabling lower overall flow [8,12]. To the best of our knowledge, this marks the first “ex vivo” study directly comparing this specific irrigation technology with a current standard 4 mm non-irrigated catheter, thereby revealing a decrease in the superficial lesion areas using this unique irrigation type. This irrigation method proficiently reduces the temperature of the endocardial surface, thereby leading to a myocardial lesion that is notably more precise and featuring a diminished surface area and an increased depth only at higher power settings (40 W) when compared with a non-irrigated catheter. This discovery implies that this technology may be particularly advantageous when targeting smaller areas, thereby minimizing the risks of collateral damage. Nowadays, irrigated catheters, including FlexAbility^TM^, are widely utilized for atrial flutter, atrial fibrillation, and ventricular ectopia/tachycardia ablation [13]. The approach to atrioventricular nodal reentry tachycardia (AVNRT) ablation using irrigated tip catheters is not widespread throughout ablation centers worldwide. The use of irrigated catheters for AVNRT ablation rose from 0% in 2005 to 23% in 2015 but remains low [14]. These data might reflect a diffuse safety procedure concerning their use. Nevertheless, at present, there are promising studies in the literature regarding the use of irrigated catheters in AVNRT ablation, demonstrating the safety and efficacy of the procedure [4,15]. Notably, to minimize the risk of iatrogenic conduction disturbance when performing AVNRT ablation, it is crucial to avoid radiofrequency delivery in the region of the compact atrioventricular (AV) node. Therefore, gross and histological dissections showed that the compact AV node is superficial, with a distance from the right-sided endocardium of a few millimeters [16]. The precise ablation of the target is essential to avoid the involvement of the compact AV node. Our data suggest that the superficial length, width, and area of the lesions performed using the non-irrigated catheter were significantly higher; moreover, lesions made using the irrigated ablation catheter were only deeper at a high-power delivery. This might support the idea that an irrigated ablation catheter, which is commonly matched with a 3D electroanatomic mapping (3D EAM) system, might allow for the more precise targeting of the slow pathway, thereby reducing the risk of affecting the compact AV node. Finally, our data show that an irrigated tip catheter might minimize the risk of steam pops, which can be dangerous in the triangle of Koch, thereby determining a larger lesion and potentially compromising the AV node [8]. 

## 5. Limitations

The findings of this “ex vivo” study were attained using pig heart slices, offering a controlled environment for precise catheter orientation and contact. While this model ensures reproducibility and advantageous experimental conditions, it is important to note that the results may not perfectly replicate the myocardial lesions in a beating heart. In the dynamic environment of a beating heart, delivering a RF current to the endocardium can lead to irregular lesion shapes and an increased variability in lesion size. Quantifying the impact of different power deliveries using the two catheters may be more challenging in such circumstances. Furthermore, disparities in lesion dimensions between the heart slice and the beating heart may arise due to the absence of blood supply during heart contractions. Blood flow in a beating heart could potentially dissipate heat during RF application, thereby influencing the resulting lesion size and steam pop incidence.

## 6. Conclusions

This study highlights that RF ablation using an irrigated flexible-tip catheter has the potential to generate smaller superficial lesion areas compared with those generated using a non-irrigated catheter. Furthermore, the use of this catheter may contribute to a decrease in steam pops even with escalating power output. Our findings indicate that an irrigated catheter could be a preferable choice when precision is crucial, particularly in determining the superficial length, width, and area of lesions (Appendix A).

## Figures and Tables

**Figure 1 biology-13-00132-f001:**
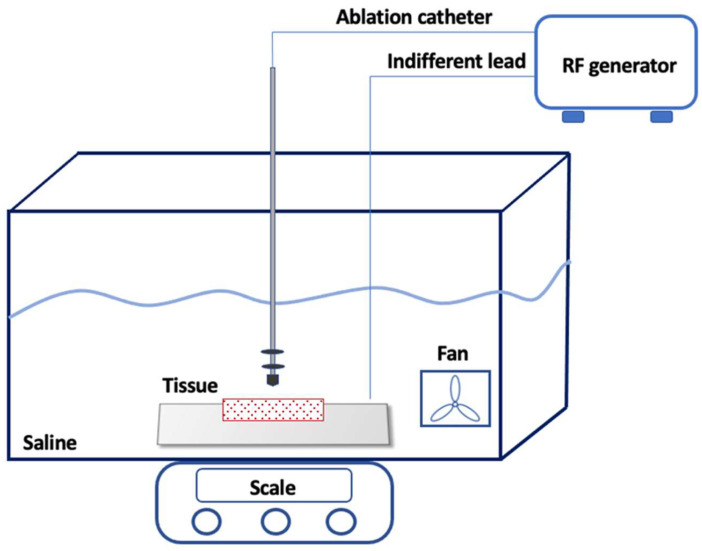
Experimental Setup. The ablation catheter was carefully placed perpendicularly to the myocardial surface using a manipulator. A high-precision digital scale was placed under the chamber. Inside the saline chamber, a pump generated the non-pulsatile flow towards the ablation electrode–tissue interface. The ablation radiofrequency generator continuously monitored the applied voltage, current, power (1 to 100 W), impedance (50 to 300 Ohms), and temperature (set at 60 °C for the 4 mm catheter and 43 °C for the irrigated catheter). The current was applied in a unipolar mode between the distal electrode of the catheter and an indifferent lead placed underneath the myocardium (though not directly on it).

**Figure 2 biology-13-00132-f002:**
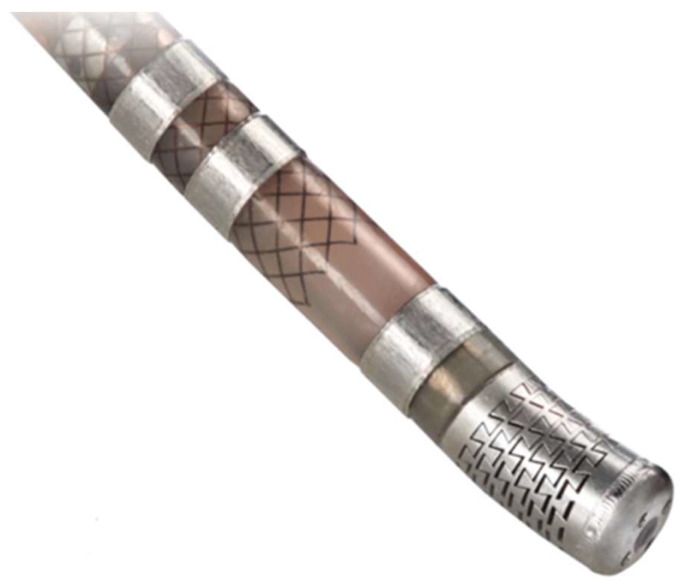
Irrigated ablation catheter with a 4 mm laser-cut flexible tip.

**Figure 3 biology-13-00132-f003:**
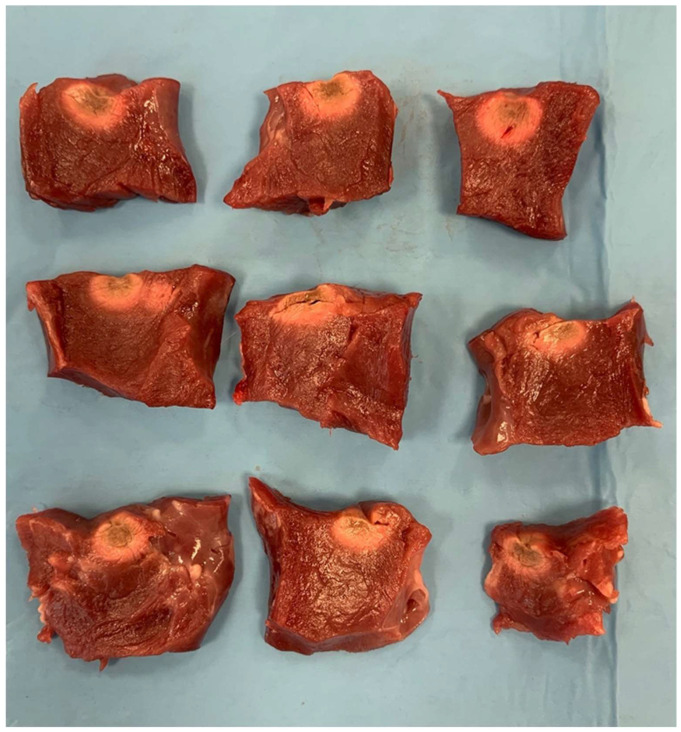
Lesion patterns with an irrigated catheter. The top row displays three slabs with RF lesions created using the irrigated catheter at 40 W. The second row shows slabs with RF lesions at 30 W, and the third row depicts slabs with RF lesions at 20 W using the irrigated catheter.

**Figure 4 biology-13-00132-f004:**
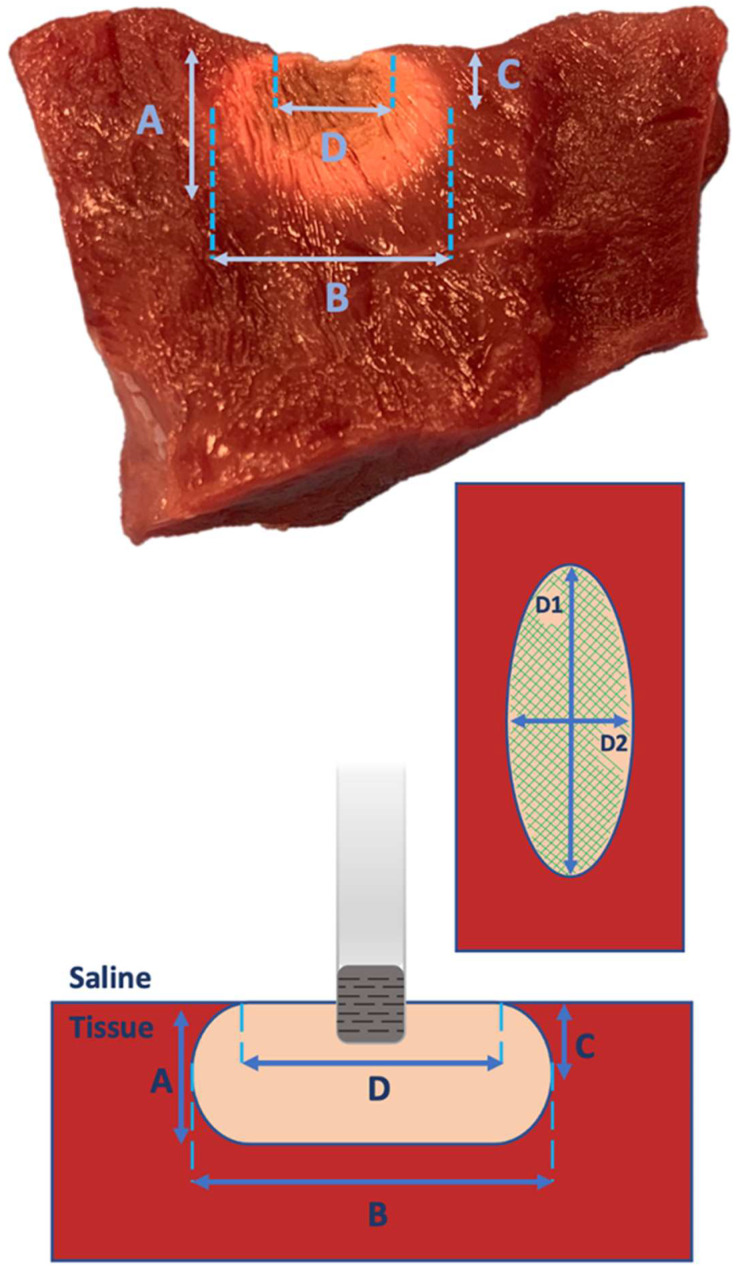
Lesion Characteristics. (A) Lesion depth, (B) maximum length of the lesion at any depth, (C) depth at the maximum length, and (D) lesion surface diameters (length and width: D1 and D2). In the upper and lower parts of the image, cross-sections of the RF lesion are depicted; in the center to the right, the endocardial superficial scheme of the lesion is shown with the highlighted length (D1), width (D2), and superficial area (green lines).

**Figure 5 biology-13-00132-f005:**
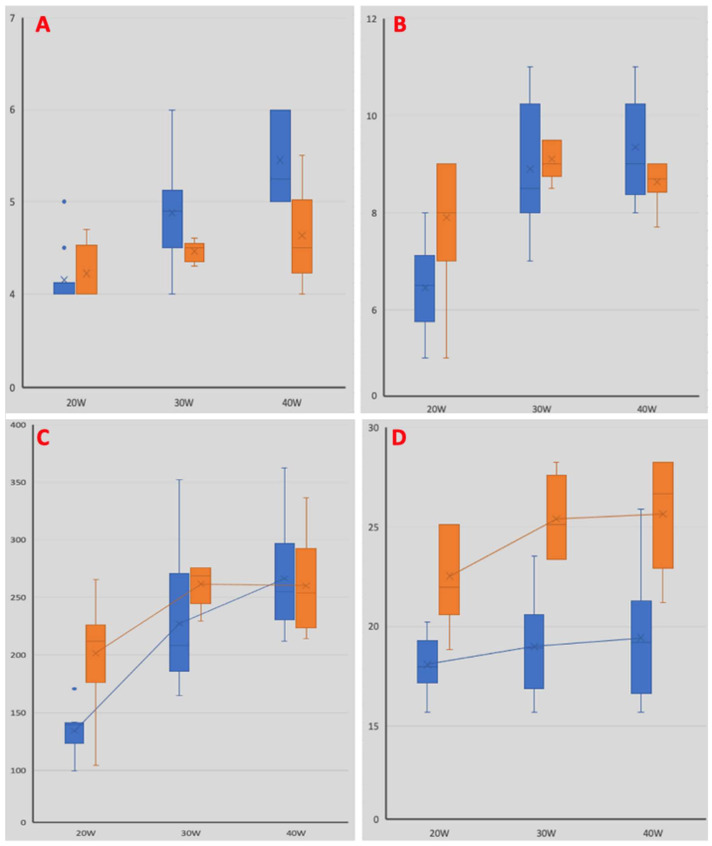
Comparison of lesion characteristics. Blue bars depict lesion characteristics obtained using the irrigated catheter, while orange bars represent those obtained using the non-irrigated catheter. Panel (**A**): histograms illustrating the maximum lesion depth at different power levels. Panel (**B**): histograms illustrating the maximum lesion length at different power levels. Panel (**C**): histograms illustrating the maximum lesion volumes at different power levels. Panel (**D**): histograms illustrating the maximum lesion superficial area at different power levels.

**Table 1 biology-13-00132-t001:** Lesion characteristics at 20 W.

20 W	Irrigated Catheter	Non-Irrigated Catheter	*p*
Superficial length [mm]	5.3 ± 0.3	7.7 ± 0.8	0.02
Superficial width [mm]	4.4 ± 0.3	3.8 ± 0.4	0.001
Maximum depth [mm]	4.2 ± 0.3	4.2 ± 0.3	0.43
Maximum length at any depth [mm]	6.5 ± 1	7.9 ± 1.2	0.012
Depth at max length [mm]	2.1 ± 0.4	2.3 ± 0.4	0.72
Superficial area [mm^2^]	18 ± 1.5	23 ± 2.5	0.001
Lesion volume [µL]	134 ± 18	201 ± 42	0.002

**Table 2 biology-13-00132-t002:** Lesion characteristics at 30 W.

30 W	Irrigated Catheter	Non-Irrigated Catheter	*p*
Superficial length [mm]	5.5 ± 0.4	8.5 ± 0.3	0.03
Superficial width [mm]	4.4 ± 0.5	3.9 ± 0.2	0.019
Maximum depth [mm]	4.9 ± 0.5	4.5 ± 0.1	0.08
Maximum length at any depth [mm]	8.9 ± 1.3	9.1 ± 0.4	0.45
Depth at max length [mm]	2.1 ± 0.5	2.2 ± 0.4	0.98
Superficial area [mm^2^]	19 ± 2.4	25 ± 4.6	0.005
Lesion volume [µL]	227 ± 58	131 ± 32	0.86

**Table 3 biology-13-00132-t003:** Lesion characteristics at 40 W.

40 W	Irrigated Catheter	Non-Irrigated Catheter	*p*
Superficial length [mm]	5.5 ± 0.4	8.6 ± 0.4	0.02
Superficial width [mm]	4.5 ± 0.5	3.8 ± 0.3	0.003
Maximum depth [mm]	5.5 ± 0.5	4.6 ± 0.5	0.007
Maximum length at any depth [mm]	9.4 ± 1.1	8.6 ± 0.4	0.14
Depth at max length [mm]	2.5 ± 0.4	2.4 ± 0.4	1
Superficial area [mm^2^]	19 ± 3.2	26 ± 2.6	0.001
Lesion volume [µL]	266 ± 43	260 ± 39	0.73

## Data Availability

The experimental data used to support the findings of this study are available from the corresponding author upon request.

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
