# Peer review of "Comparative Study of Lesions Obtained through Radiofrequency between the Irrigated Ablation Catheter with a Flexible Tip and the Non-Irrigated Catheter in Ex Vivo Porcine Hearts"

_biology, 2024, doi:10.3390/biology13020132_

Round 1

Reviewer 1 Report

Comments and Suggestions for Authors

The study "Comparative study of lesions obtained with radiofrequency between irrigated ablation catheter with flexible tip and non-irrigated catheter in ex vivo porcine hearts" presents a thorough comparison of lesion characteristics created by different types of radiofrequency ablation catheters.

The research demonstrates that irrigated catheters with flexible tips can generate lesions with smaller superficial areas compared to non-irrigated catheters. Additionally, these catheters could potentially create deeper lesions at higher power settings, offering a more precise approach for targeting specific myocardial areas. This research adds valuable insights into the comparative effectiveness of different catheter types, with implications for improving procedural outcomes in cardiac ablation therapies. Do the authors plan to test IRE catheters in the ex-vivo setup?

Author Response

We appreciate the comments provided by reviewer 1. However, it's important to note that our study focused exclusively on radiofrequency energy. In future research, we plan to explore the testing of linear IRE catheters

Reviewer 2 Report

Comments and Suggestions for Authors

Summary:
This is a very well-formulated and executed study where the authors sought to explore the lesional characteristics generated by an irrigated, flexible tip ablation catheter compared with those by a non-irrigated system in porcine heart slabs. While some literature in the clinical realm compares these systems, this is indeed the first study evaluating this particular irrigated system in an ex-vivo environment. The authors have found that irrigated-tip catheters may likely generate a comparatively more precise lesion compared to their non-irrigated counterparts with increased depth and fewer incidences of steam pops even at higher energy outputs. 

The study is well-designed and executed. The methodology and statistical descriptions are clear and succinct. The discussion is adequate and important limitations have been acknowledged. This study certainly contributes additional important data to the field and will allow readers and especially operators a deeper understanding of lesional characteristics produced by these systems.

Major comments:
None.

Minor comments: 
- Consider including images of the lesion patterns with the non-irrigated catheters as well, if possible, for comparison. 
- The description in various areas of the manuscript regarding the superficial lesions created by both systems is a little confusing concerning the length, width, and area of the lesion. While this becomes clearer when looking at Figure 4 showing the lesion characteristics, the description itself can use further clarification to avoid any confusion. 

Author Response

We thank reviewer 2 for the comments.

Macroscopic images of the lesion patterns with the non-irrigated catheters are very similar to the ones obtained with the irrigated ones. We thought it did not add great value to understanding the differences between the two types of lesions. We added it here as supplemental for reviewer 2 (from left to right 20W, 30W, 40W), and if he/she suggests adding the image, we will surely include it in the paper

As suggested, we modified the text using the description of Figure 4 to enhance the clarity of the different lesion measures from line 130 to line 135: "Using a digital precision caliper, we assessed lesion depth, (B) maximum length of the lesion at any depth, (C) Depth at the maximum length, and (D) Lesion surface diameters (length and width: D1 and D2) (Figure 4). We assessed superficial length (D1) defined as the long axis of the imprint left by the catheter tip in the tissue, and superficial width (D2) perpendicularly to the previous axe. Subsequently, the superficial area of each lesion was calculated (Figure 4). "

Reviewer 3 Report

Comments and Suggestions for Authors

This work from an academic cardiology center compares biological adaptation in myocardial tissues treated by either flexible tip or non-irrigated solid tip catheters. The authors employed an experimental setting previously validated by another research group using a chamber filled of saline solution under stable temperature and continuous weighting for assessing contact force.

The study tested porcine cardiac samples. The biophysics setting showed greater superficial lesion areas when the non-irrigated catheter was used compared with the irrigated catheter, and the irrigated catheter determined significantly greater values of lesion maximum depth; the calculated volume of the lesions performed with the non-irrigated catheter was significantly greater at low power delivery; no steam pop was observed using the irrigated ablation catheter, while 4 steam pops occurred with the non-irrigated catheter delivering 30 W, and 8 steam pops occurred delivering 40W.

The work appears methodologically correct although a documentation of steam pop formation and embolization is the only technical aspect that is missing in this report; either a movie file, a direct picture, or a cartoon could help in differentiating the worst outcome that derives from the transcatheter ablation techniques.

Comments on the Quality of English Language

english language is fine

Author Response

We thank reviewer 3 for the comments. Steam pops were visually registered during RF application, while carbonization was assessed during lesion analysis. We have video documentation of steam pops, but the quality is low, so we do not think it is valuable for publication. If it is needed we can provide it to reviewer 3. 

We modify the text to make it more clear from line 120 to line 130: 

"During each RF energy application, the delivered power, the impedance, and the temperature at the tip electrode were continuously recorded. The occurrence of an audible steam pop was noted and visually confirmed, and the lesion formed by steam pop has been excluded from the count, but the number of steam pops in every session were collected.

2.4. Lesion measurements and characterization

Following the administration of radiofrequency energy, the myocardium was cross sectioned at the level of each lesion (Figure 3). All lesions underwent examination for endocardial surface carbonization and crater formation, defined as the disruption of myocardial tissue. Carbonization manifested as superficial tissue damage with indications of endocardial surface burning"

Round 2

Reviewer 3 Report

Comments and Suggestions for Authors

The authors fulfilled the requested modifications

Comments on the Quality of English Language

Thank you for consideration of my reviewer activity.

best regards,

Antonio Curcio

Chief, Division of Cardiology

UNICAL

Rende, Cosenza

ITALY

Author Response

Thank you for your useful comments that improve the overall quality of the work. 

Best regards

Francesco Vitali, MD, PhD